# Single-cell RNA sequencing of peripheral blood mononuclear cells from acute Kawasaki disease patients

Zhen Wang [1,11✉], Lijian Xie[2,11], Guohui Ding [3,4,11], Sirui Song[2], Liqin Chen[2], Guang Li [5], Min Xia[6], Dingding Han[6], Yue Zheng[6], Jia Liu[5], Tingting Xiao[2], Hong Zhang[6], Yujuan Huang[2], Yixue Li [7,8,9,10 ✉] & Min Huang [2✉]

Kawasaki disease (KD) is the most common cause of acquired heart disease in children in developed countries. Although functional and phenotypic changes of immune cells have been reported, a global understanding of immune responses underlying acute KD is unclear. Here, using single-cell RNA sequencing, we profile peripheral blood mononuclear cells from seven patients with acute KD before and after intravenous immunoglobulin therapy and from three age-matched healthy controls. The most differentially expressed genes are identified in monocytes, with high expression of pro-inflammatory mediators, immunoglobulin receptors and low expression of MHC class II genes in acute KD. Single-cell RNA sequencing and flow cytometry analyses, of cells from an additional 16 KD patients, show that although the percentage of total B cells is substantially decreased after therapy, the percentage of plasma cells among the B cells is significantly increased. The percentage of CD8[+] T cells is decreased in acute KD, notably effector memory CD8[+] T cells compared with healthy controls. Oligoclonal expansions of both B cell receptors and T cell receptors are observed after therapy. We identify biological processes potentially underlying the changes of each cell type. The single-cell landscape of both innate and adaptive immune responses provides insights into pathogenesis and therapy of KD.

[1] Shanghai Institute of Nutrition and Health, University of Chinese Academy of Sciences, Chinese Academy of Sciences, Shanghai, China. [2] Department of Cardiology, Shanghai Children's Hospital, Shanghai Jiaotong University, Shanghai, China. [3] Institute for Digital Health, International Human Phenome Institutes (Shanghai), Shanghai, China. [4] Gui'an Bio-Med Big Data Center, Shanghai Institute of Nutrition and Health, Chinese Academy of Sciences, Guiyang, China. [5] Shanghai QianBei Med. Technology Co. Ltd, Shanghai, China. [6] Department of Clinical Laboratory, Shanghai Children's Hospital, Shanghai Jiaotong University, Shanghai, China. [7] Hangzhou Institute for Advanced Study, University of Chinese Academy of Sciences, Hangzhou, China. [8] Bio-Med Big Data Center, Shanghai Institute of Nutrition and Health, Chinese Academy of Sciences, Shanghai, China. [9] Collaborative Innovation Center for Genetics and Development, Fudan University, Shanghai, China. [10] Guangzhou Laboratory, Guangzhou, China. [11]These authors contributed equally: Zhen Wang, Lijian Xie, Guohui Ding. ✉email: zwang01@sibs.ac.cn; yxli@sibs.ac.cn; huangmin@sjtu.edu.cn

Kawasaki disease (KD) is an acute, systemic febrile illness and vasculitis of childhood that can lead to coronary artery lesions (CALs)[1]. This disease predominantly affects children who are younger than 5 years and has become the most common cause of acquired heart disease among children in many developed countries[2]. The etiology of KD remains elusive, and diagnosis continues to depend on principal clinical features including fever, rash, conjunctivitis, changes in the oral mucosa and in the extremities, and cervical lymphadenopathy[3]. As the signs and symptoms of KD resemble those of other childhood febrile illnesses, prompt diagnosis of KD is still challenging. High-dose intravenous immunoglobulin (IVIG) within the first 10 days after fever onset is the standard therapy for KD, which remarkably reduces the rate of CALs[4]. However, the mechanism of IVIG in the treatment of KD is unknown. Approximately 10% to 20% of KD patients are IVIG-resistant and are at increased risk for CALs[3]. Recently, a new multisystem inflammatory syndrome in children (MIS-C) has been reported during the severe acute respiratory syndrome coronavirus 2 (SARS-CoV-2) epidemic[5,6]. Despite of different epidemiology and clinical characteristics between MIS-C and KD, persistent fever and common mucocutaneous findings are present in both of the disorders[7,8]. Therefore, it is urgently needed to distinguish MIS-C and KD in research investigations[9].

The most widely favored theory of KD etiology is that it is caused by a ubiquitous infectious agent in a small subset of genetically predisposed children[10]. Genome-wide wide association studies (GWAS) have identified a number of susceptibility genes for KD[11]. Many of the susceptibility genes including *ITPKC*, *CASP3*, *FCGR2A*, *CD40*, and MHC class II are related to immune functions. Nonetheless, the causative agents initiating the disease have been still widely debated over 50 years. Various pathogens have been proposed as the trigger including superantigen toxin, Epstein–Barr virus, coronavirus and retrovirus, but none has been confirmed by subsequent studies[12]. Despite the uncertainty of the cause, activation of the immune system provides important evidence for its pathogenesis. Plasma levels of pro-inflammatory cytokines such as tumor necrosis factor (TNF), interleukin (IL)-1β, and interferon (IFN)-γ are elevated during the acute phase of KD[13]. Innate immune cells including neutrophils and monocytes are elevated in the peripheral blood and have been identified in the arterial wall early in the disease[14]. Activation of peripheral blood lymphocytes seems also responsible for the development of KD, though their roles remain controversial[15]. In addition, antigen-specific IgA plasma cells and CD8[+] T cells infiltrate inflamed tissues, implying an immune response to an intracellular pathogen entering through the respiratory tract[16].

The immune system is composed of numerous cell types with varying states. Most previous studies performed immunophenotyping based on low-throughput assays, which were constrained to a few selected cell types and markers. Several studies performed transcriptome analyses of KD on bulk cell populations[17,18], but the heterogeneity of the immune system cannot be resolved. Over the past few years, the revolution in single-cell RNA sequencing (scRNA-seq) has enabled an unbiased quantification of gene expression in thousands of individual cells, which provides an more efficient tool to decipher the immune system in human diseases[19].

In this work, we profile peripheral blood mononuclear cells (PBMCs) isolated from acute KD patients at single-cell resolution. We show that monocytes are the major source of pro-inflammatory mediators and could be a therapeutic target in PBMCs. Plasma cells and effector memory CD8[+] T cells are also involved in KD, which have oligoclonal expansion of B-cell receptors (BCRs) and T-cell receptors (TCRs) after IVIG therapy. Our results show global and dynamic immune responses unique to each cell type in the pathogenesis and therapy of KD.

## Results

**Study design and single-cell RNA profiling of PBMCs.** We collected 14 fresh peripheral blood samples derived from seven patients with acute KD (P1–P7, Supplementary Table 1). The patients were diagnosed according to the criteria proposed by the American Heart Association[3]. Six patients met criteria for complete KD and one (P3) for incomplete KD. For each patient, the first blood sample was taken on the 5th days after the onset of fever before IVIG therapy. The second sample was obtained at 24 h after completion of IVIG therapy and subsidence of fever (Supplementary Table 2). All patients responded to IVIG treatment and had not developed CALs. We also collected fresh peripheral blood samples from three age-matched healthy donors as controls (H1–H3, Supplementary Table 1).

We used the 10× Genomics platform for scRNA-seq of PBMCs isolated from the samples. About 12,000 PBMCs per sample were loaded onto the platform (Supplementary Table 3) and about 6000 cells per sample could be recovered from the sequencing data (Supplementary Table 4). The total number of detected cells passing quality control was 84,986, including 34,073 cells from patients before IVIG therapy, 36,225 cells after therapy, and 14,688 cells from healthy controls (Supplementary Table 4). We also performed single-cell B-cell receptor sequencing (scBCR-seq) and single-cell T-cell receptor sequencing (scTCR-seq) based on the scRNA-seq libraries. Totally, 23,044 and 48,456 cells with productive paired BCRs and TCRs were detected, respectively (Supplementary Tables 5 and 6).

Based on the scRNA-seq profiles, we clustered the cells across samples with Seurat[20] and visualized them in two-dimensional space (Fig. 1). The cell clusters were annotated with SingleR[21] (Supplementary Fig. 1) and refined with expression of canonical marker genes. Major cell types comprising PBMCs could be well captured by scRNA-seq (Supplementary Fig. 2), including T cells (*CD3D*, *CD3E*, *CD3G*, 60.00%), CD4[+] T cells (*CD4*, 36.94%), CD8[+] T cells (*CD8A*, *CD8B*, 18.53%), natural killer (NK) cells (*NCAM1* or *CD56*, *KLRB1*, *NKG7*, 5.44%), B cells (*CD19*, *MS4A1* or *CD20*, *CD38*, 22.27%), monocytes (*CD14*, *CD68*, *FCGR3A* or *CD16*, 9.60%), myeloid dendritic cells (mDCs) (*CD1C*, 0.26%), plasmacytoid dendritic cells (pDCs) (*LILRA4*, 0.18%) and hematopoietic stem and progenitor cells (HSPCs) (*CD34*, 0.12%). There were also some residual erythrocytes (*HBB*, 0.99%) and megakaryocytes (*PPBP*, 0.92%) mixed in the PBMCs. We pooled the PBMCs by conditions, and found that compared with healthy controls, pre-treatment KD patients showed an increased percentage of monocytes and B cells, while a decreased percentages of T cells and NK cells (Fig. 1 and Supplementary Fig. 3). These observations were in line with previous clinical reports[15,22], suggesting that scRNA-seq recovered key immune alterations of KD.

We compared the scRNA-seq data with a public bulk gene expression dataset (GSE73577), which profiled PBMCs from 19 KD patients before and after IVIG treatment with cDNA microarray[23] (Supplementary Fig. 4). The fold-changes across all genes were significantly positively correlated between the two datasets (Pearson's $R = 0.34$, $P < 2.2 \times 10^{-16}$), and the correlation became much stronger for 1054 differentially expressed genes (DEGs, false-discovery rate [FDR] <0.05) in both of the datasets (Pearson's $R = 0.74$, $P < 2.2 \times 10^{-16}$). These results demonstrated consistent global gene expression patterns between the two datasets.

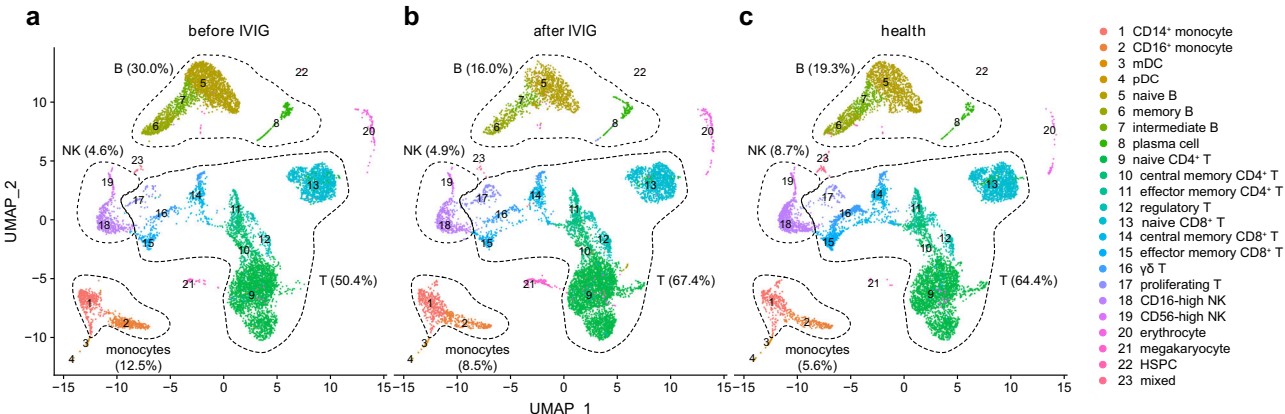

**Fig. 1 Two-dimensional representation of cell types identified by scRNA-seq.** The dimensional reduction is performed with the uniform manifold approximation and projection (UMAP). Each dot represents a cell, which is colored according to cell type. The cells are pooled across all patients and separated by conditions: **a** KD before intravenous immunoglobulin (IVIG) therapy; **b** KD after IVIG therapy; **c** healthy controls. Percentages of major cell types under each condition are shown on the plot, and percentages of subtypes are shown on Supplementary Fig. 3. mDC, myeloid dendritic cells; pDC, plasmacytoid dendritic cells; HSPC, hematopoietic stem and progenitor cells. Source data are provided in the Source Data file.

**Dynamics of major PBMC immune cell types**. We compared the percentage of each immune cell type for KD patients before and after IVIG therapy, as well as healthy controls on the individual level (Fig. 2a). In parallel to scRNA-seq, we also performed flow cytometric analysis of peripheral blood lymphocytes from 16 additional complete KD patients and 20 age-matched healthy donors (Fig. 2b). The scRNA-seq data demonstrated that KD patients before therapy had an increased percentage of B cells ($P = 0.02$, two-sided $t$-test) and a decreased percentage of CD4$^+$ T cells ($P = 0.01$, two-sided $t$-test) compared with those after therapy (Fig. 2a). These results were similar to most previous studies[15,22,24,25], and could be confirmed by our flow cytometric analysis (Fig. 2b). Moreover, the percentage of B cells was substantially increased in KD patients before therapy compared with healthy controls ($P = 5.28 \times 10^{-6}$, two-sided $t$-test), which quickly returned to normal after therapy (Fig. 2b). The scRNA-seq data also showed a lower percentage of CD8$^+$ T cells in KD patients than healthy controls ($P = 0.02$, two-sided $t$-test, Fig. 2a), which was again consistent with previous reports[15,22,24,25] and our flow cytometric results ($P = 2.68 \times 10^{-3}$, two-sided $t$-test, Fig. 2b). A reduction of NK cells was reported in acute KD[22,26,27] and can be verified by our flow cytometric dataset ($P = 0.04$, two-sided $t$-test, Fig. 2b). The similar tendency of NK cells could be observed with the scRNA-seq data, though statistical significance was lacking due to the smaller sample size for scRNA-seq (Fig. 2a).

It was well established that the abundance of monocytes was decreased following IVIG therapy[15,28,29]. However, one patient (P1) in the scRNA-seq data showed a remarkably increased percentage of monocytes after therapy (Fig. 2a). To exclude any potential technical bias, we checked the routine blood test data of the patients for scRNA-seq and flow cytometry (Fig. 2c). The data confirmed a significant decrease of monocytes after therapy in general ($P = 6.26 \times 10^{-5}$, two-sided $t$-test), whereas only P1 displayed a reverse tendency similar to that found with scRNA-seq.

**Monocyte subsets and responses**. Next, we considered the subsets of each cell type to dissect their heterogeneity. Two subsets of monocytes were recognized, including CD14$^+$ monocytes (*CD14*, 70.30%) and CD16$^+$ monocytes (*FCGR3A*,29.70%) (Fig. 1 and Supplementary Figure 5). After IVIG therapy, the patients showed a significantly decreased percentage of CD16$^+$ monocytes ($P = 0.02$, two-sided $t$-test, Fig. 3a and Supplementary Fig. 6). It

was reported that CD16$^+$ monocytes were more closely associated with many inflammatory conditions[30], including acute KD[15,29]. The percentage of CD16$^+$ monocytes remained significantly lower in post-treatment KD patients than healthy controls ($P = 1.96 \times 10^{-3}$, two-sided $t$-test), but we did not find significant difference between pre-treatment KD patients and healthy controls (Fig. 3a).

To identify more genes involved in the immune responses, we performed differential expression analysis separately for each cell type using DESeq2[31] (Supplementary Data 1). The most DEGs (FDR < 0.05) resulted from monocytes among all PBMC populations (Supplementary Figure 7). Gene ontology (GO) and pathway over-representation analyses of the DEGs (Supplementary Data 2) suggested that they were significantly enriched in S100 proteins, immunoglobulin receptors, cytokine receptors, complement and receptors and MHC class II receptors (FDR < 0.05, Fig. 3b). *S100A8*, *S100A9* and *S100A12* are predominantly expressed by neutrophils and monocytes, which exert pro-inflammatory functions in a range of diseases[32]. Consistent with previous findings[33,34], these genes were significantly overexpressed before therapy compared with those after therapy and healthy controls (Fig. 3b). Similarly, most DEGs encoding immunoglobulin receptors were overexpressed in pre-treatment KD patients, including *FCGR3A*, *FCGR3B*, *FCGR2A*, *FCGR2B*, *FCGR1B* and *FCEG1G* (Fig. 3b). Notably, although *FCGR2B* was considered as an inhibitory receptor in immune complex driven reactions[35], our results were coherent to a previous study that the expression of *FCGR2B* in monocytes was reduced after IVIG therapy[36]. In addition, we found that most DEGs of complement and receptors were upregulated before therapy, including *C1QA*, *C1QB*, *C1QC*, *C2*, *CFD*, *C3AR1* and *C5AR1* (Fig. 3b). In contrast, most DEGs encoding MHC class II receptors were downregulated in KD patients compared to healthy controls regardless of therapy, including *HLA-DQA1*, *HLA-DQB1*, *HLA-DRA*, *HLA-DPB1*, *HLA-DMA*, *HLA-DMB*, and *HLA-DOA* (Fig. 3b). The expression patterns of DEGs encoding cytokine and chemokine receptors were more diverse. While some of them were overexpressed in pre-treatment patients, including *IFNAR2*, *IFNGR2*, *IL3RA*, *IL10RB*, and *CX3CR1*, others were overexpressed in post-treatment patients or in healthy controls, including *IL7R*, *IL13RA1*, *IL6ST*, and *CXCR3* (Fig. 3b).

As hypercytokinemia is one of the most important characteristics of KD, we traced the expression of cytokines reported to be elevated in KD[13] (Fig. 3c). IL-1β and TNF are crucial

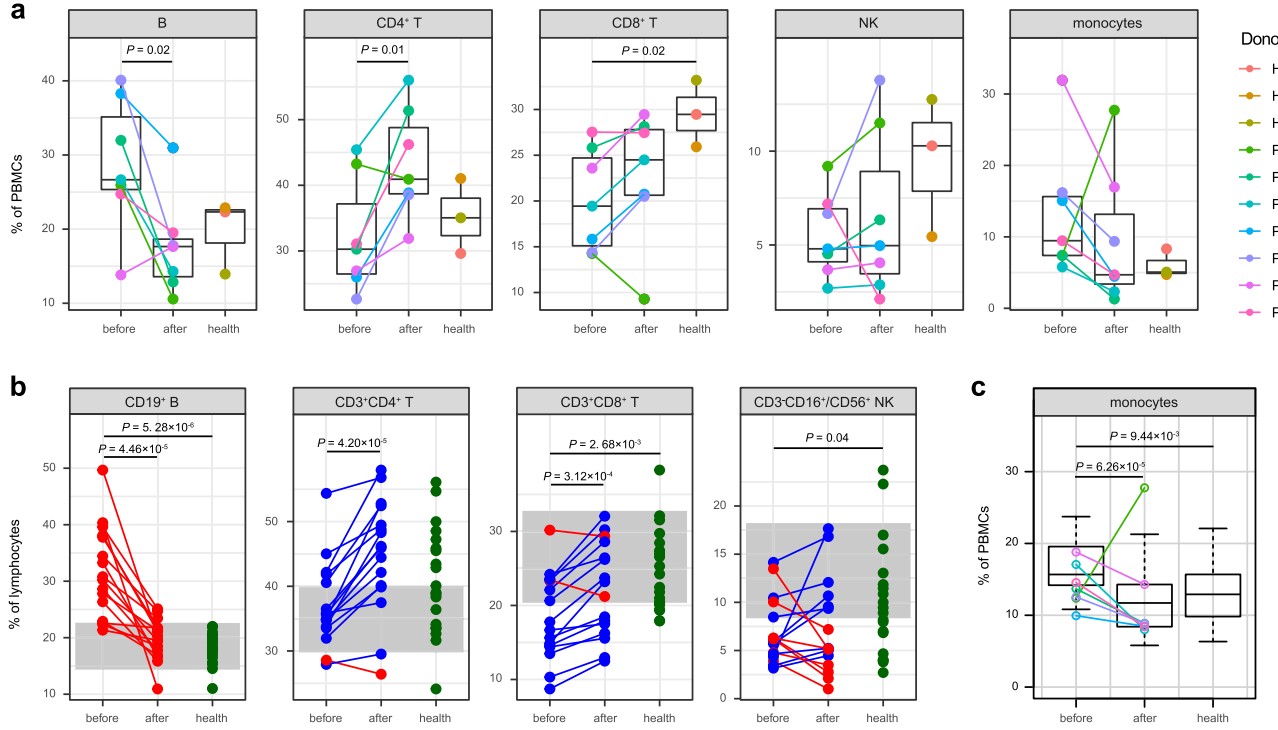

**Fig. 2 Comparison of major cell types across conditions. a** Percentage of each cell type in PBMCs revealed by scRNA-seq. The samples are colored according to donors. *P*-values are calculated between pre- and post-treatment KD patients (*n* = 7) and healthy controls (*n* = 3) by using the two-sided *t*-test. **b** Percentage of lymphocyte populations by flow cytometric validation on additional KD patients (*n* = 16) and healthy controls (*n* = 20). Patients with a decreased percentage after therapy are colored in red, and patients with an increased percentage after therapy are in blue. Healthy controls are colored in green. The gray areas represent the reference ranges of the panel. *P*-values between conditions are calculated by using the two-sided *t*-test. **c** Percentage of monocytes by routine blood test of samples collected for both scRNA-seq (KD patients *n* = 7) and flow cytometric analysis (KD patients *n* = 16, healthy controls *n* = 20). KD patients for scRNA-seq are highlighted with the same color as (**a**). *P*-values are calculated by using the two-sided *t*-test. In (**a**) and (**c**), data are represented as boxplots where the middle line is the median, the lower and upper hinges correspond to the first and third quartiles, and the whiskers extend from the hinge to the farthest data point within a maximum of 1.5 × interquartile range. Source data are provided in the Source Data file.

inflammatory mediators during the pathogenesis of KD and have been used as therapeutic targets to treat IVIG-resistant patients[37]. We found that both *IL1B* and *TNF* were mainly expressed in monocytes among PBMCs and were upregulated in pre-treatment KD patients (Fig. 3d). In addition to S100 genes, monocytes were also the primary source to express *CCL2*, *CXCL10*, *CXCL8*, *HGF*, *IL15*, *IL18*, *RETN* and *TNFRSF1B* (Fig. 3c). Although not all of them could be detected as DEGs across conditions, the total production of the cytokines were expected to be elevated in pre-treatment KD patients with the increased abundance of monocytes.

**B-cell subsets and BCR repertoires**. We identified four distinct B-cell subsets (Fig. 1 and Supplementary Fig. 8)[38]: naive B cells (*MS4A1*, *TCL1A*, 62.73%), memory B cells (*MS4A1*, *CD27*, *IGHA1*, *IGHG1*, 17.27%), plasma cells (*CD27*, *CD38*, *IGHA1*, *IGHG1*, 6.24%) and those intermediate between naive and memory cells (13.77%). After IVIG therapy, the proportion of plasma cells in total B cells was increased compared with that before therapy (*P* = 0.02, two-sided *t*-test, Fig. 4a and Supplementary Fig. 9). As plasma cells are the terminally differentiated and antibody-secreting B cells, which were not investigated in any previous studies, we performed flow cytometric analysis of the subset for the additional 16 KD patients and 20 healthy controls (Fig. 4b and Supplementary Fig. 10). The analysis confirmed that the post-treatment patients had a significantly higher percentage of plasma cells than both pre-treatment patients (*P* = 0.01, two-sided *t*-test) and healthy controls (*P* = 0.03, two-sided *t*-test).

Although B cells showed the fewest DEGs among all PBMC populations (Supplementary Fig. 7), the DEGs were enriched in several interesting functional categories, including cell cycle process, interferon signaling pathway, and S100 genes (FDR < 0.05, Fig. 4c and Supplementary Data 1 and 2). Many DEGs positively associated with cell cycles were upregulated in pre-treatment KD patients compared with post-treatment patients such as *CCND2*, *RRM2*, *CENPM*, *NME1*, *CKS1B* and *MYC*, while DEGs to inhibit cell cycles were downregulated in pre-treatment patients such as *BTG2* and *CDKN1B* (Fig. 4c). These expression patterns were consistent with the substantially increased abundance of B cells before therapy. Several interferon response genes were upregulated after therapy, including *IRF1*, *STAT1*, *ITGB7*, *PARP14*, and *CD38*, a marker of cellular activation expressed by plasma cells (Fig. 4c). The expression of *S100A8*, *S100A9* and *S100A12* were higher in KD patients than in healthy controls, though no significance was found before and after therapy (Fig. 4c).

We analyzed the scBCR-seq data to explore dynamics of BCR repertoires during acute KD. The percentage of *IGHA* and *IGHG* were significantly elevated after IVIG therapy (*P* = 0.02, two-sided *t*-test, Fig. 4d and Supplementary Fig. 11), which implicated isotype switching of immunoglobulins from IgM/IgD to IgG/IgA resulted from B-cell activation. We then examined the clonality of BCRs based on unique VDJ sequences for each patient. Intriguingly, in most of the patients except P7, the percentages of clonal BCRs (clonal size ≥3) were substantially increased after therapy compared with those before therapy (Fig. 4e). The clonality of BCRs measured by the Gini coefficient was also

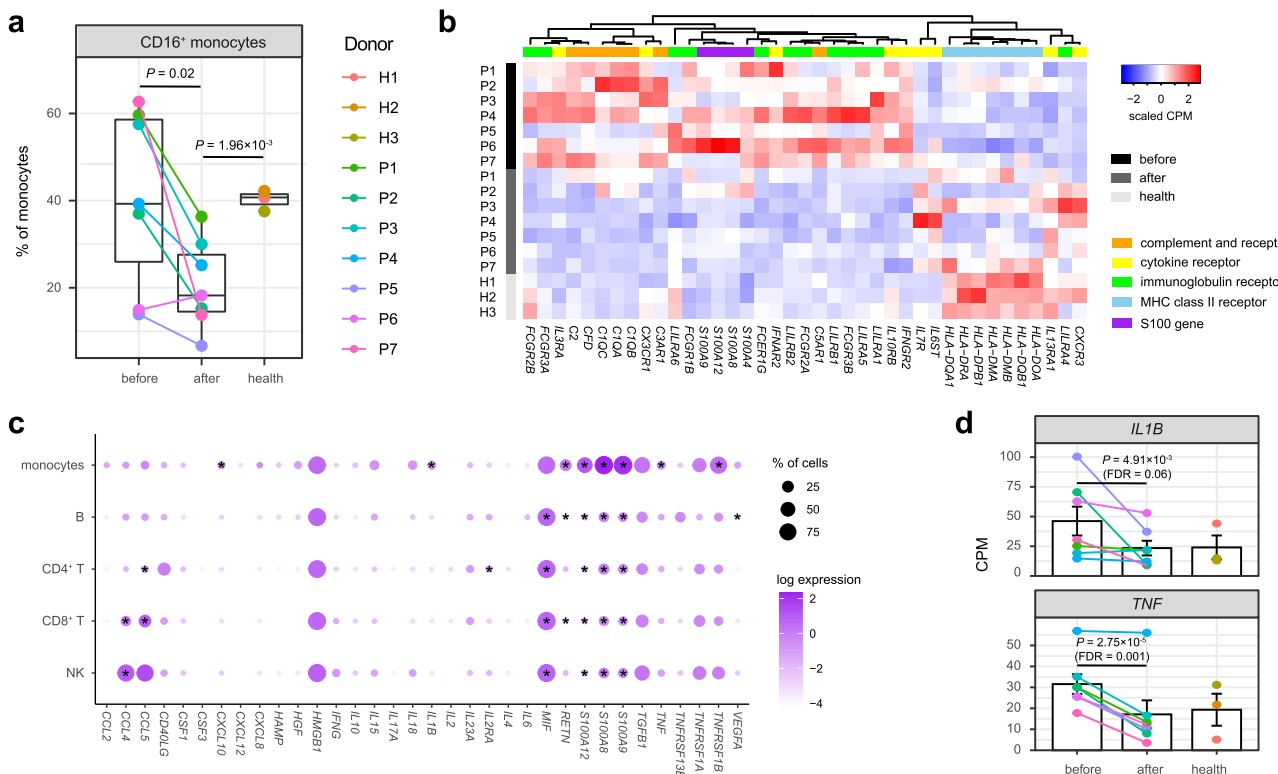

**Fig. 3 Characterization of monocytes and transcriptional patterns. a** Percentage of CD16+ monocytes across conditions recovered by scRNA-seq (KD patients $n = 7$, healthy controls $n = 3$). The middle line of the boxplot is the median, the lower and upper hinges correspond to the first and third quartiles, and the whiskers extend from the hinge to the farthest data point within a maximum of 1.5× interquartile range. The samples are colored according to donors. *P*-values are calculated by using the two-sided *t*-test. **b** Heat map of DEGs with functional enrichment in monocytes. Gene expression on the sample level is measured by the counts per million mapped reads (CPM), which is then scaled across samples. **c** Expression patterns of cytokines reported to be elevated in KD. The dot plot depicts average expression level and percentage of cells expressing the cytokine genes across PBMC populations. Cytokine genes with higher expression in pre-treatment patients than post-treatment patients or healthy controls (nominal *P*-value < 0.05 with DESeq2[31], two-sided) are marked with stars. **d** Expression level of *IL1B* and *TNF* in monocytes across conditions (KD patients $n = 7$, healthy controls $n = 3$). The bar plot depicts mean CPM and standard error of the mean. The samples are in the same colors as **a**. *P*-values and FDRs are calculated with DESeq2[31] (two-sided). Source data are provided in the Source Data file.

significantly higher in post-treatment KD patients than that in pre-treatment patients ($P = 0.03$, two-sided *t*-test) and healthy controls ($P = 0.04$, two-sided *t*-test, Supplementary Fig. 12). Particularly, in every post-treatment patient except P7, we could observe remarkable oligoclonal expansions, which were dominated by IgA and IgG isotypes (FDR < 0.01, Fisher's exact test, Fig. 4e). The *IGHV* gene usage of the expanded clones was diverse among the patients (Fig. 4e), and no obvious bias in *IGHV* usage was observed between KD patients and healthy controls (Supplementary Fig. 13), indicating specific B-cell reactions rather than responses to a superantigen.

**T- and NK-cell subsets and TCR repertoires**. We refined CD4+ T, CD8+ T, and NK-cell clusters based on expression of canonical genes, respectively (Supplementary Figs. 14 and 15). Both CD4+ T and CD8+ T cells comprised three subsets (Fig. 1): naive cells (*CCR7*, *SELL*, 70.41%), central memory cells (co-expression of *CCR7*, *SELL* and *S100A4*, 12.04%) and effector memory cells (*S100A4*, 6.54%)[39]. The memory CD8+ T cells also expressed high levels of cytotoxicity-associated genes (*GZMK*, *GZMB*). In addition, regulatory CD4+ T cells (*FOXP3*, 3.51%), γδ T cells (*TRDC*, 5.34%) and proliferating T cells (*TYMS*, 2.15%) were identified. While we did not observe significant alterations among the CD4+ T-cell subsets during acute KD (Supplementary Fig. 16), we found a significantly increased percentage of naive cells ($P = 0.01$, two-sided *t*-test) and a decreased percentage of

effector memory cells among the CD8+ T cells in KD patients compared with those in healthy controls ($P = 0.04$, two-sided *t*-test, Fig. 5a and Supplementary Fig. 17). These results indicated that the effector memory cells, which can acquire rapid cytotoxic functions following activation, were the major subset contributing to the reduction of CD8+ T cells in PBMCs. Flow cytometric analysis of the CD8+ T-cell subsets for the additional 16 KD patients (Supplementary Fig. 18) confirmed that they had a higher percentage of naive cells ($P = 0.001$, two-sided *t*-test) and a lower percentage of effector memory cells ($P = 0.03$, two-sided *t*-test) than the healthy controls, and these subsets were not significantly affected by IVIG therapy (Fig. 5b). The NK cells included two subsets (Fig. 1 and Supplementary Fig. 15): CD16-high cells (*FCGR3A*, 89.83%) and CD56-high cells (*NCAM1*, *KLRC1*, 10.17%)[40]. The percentage of CD16-high NK cells was decreased after therapy ($P = 0.03$, two-sided *t*-test, Supplementary Fig. 19), which was similar to the tendency of CD16+ monocytes.

Many DEGs in CD4+ and CD8+ T cells shared similar expression patterns related with immune responses and energy metabolisms (Supplementary Data 1 and 2). A common set of DEGs was significantly enriched in interferon signaling and responses in both CD4+ and CD8+ T cells (FDR < 0.05, Fig. 5c). While some of these DEGs were upregulated before therapy such as *IFNGR2*, *ITITM2* and *SOCS3*, others were upregulated after therapy such as *STAT1*, *IRF1*, *GBP1*, *GBP4* and *IL10RA*, a

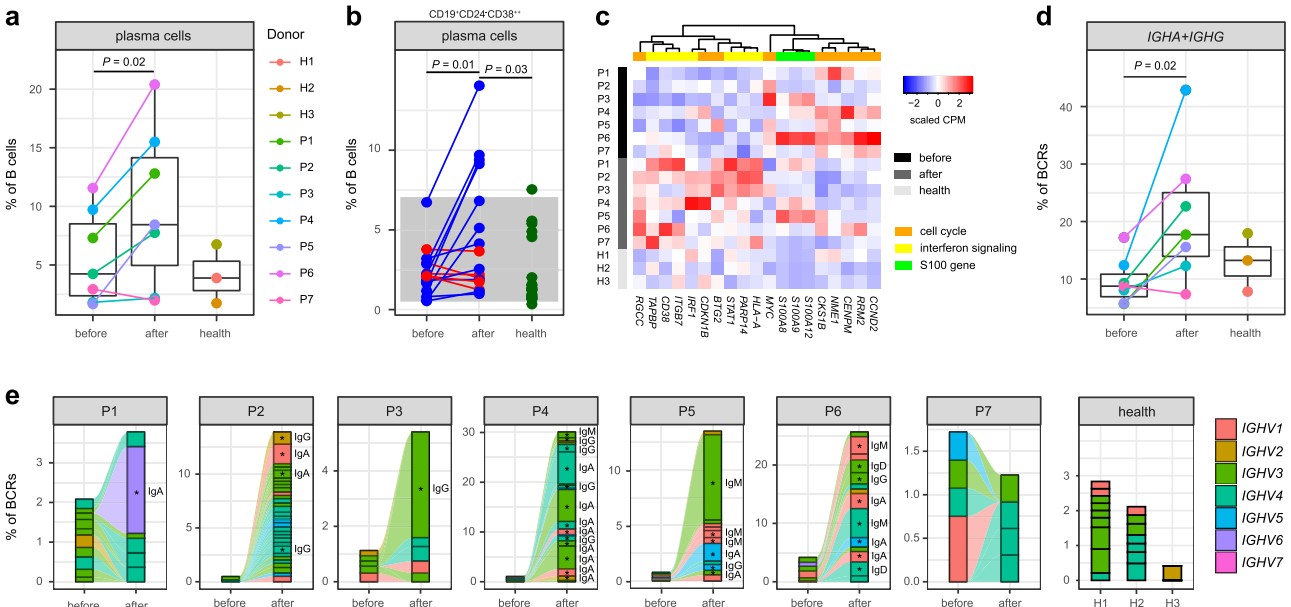

**Fig. 4 Characterization of B cells and BCRs. a** Percentage of plasma cells across conditions recovered by scRNA-seq (KD patients $n = 7$, healthy controls $n = 3$). The samples are colored according to donors. *P*-values are calculated by using the two-sided *t*-test. **b** Percentage of plasma cells (CD19+ CD24−CD38++) by flow cytometric validation on additional KD patients ($n = 16$) and healthy controls ($n = 20$). Patients with a decreased percentage after therapy are colored in red, and patients with an increased percentage after therapy are in blue. Healthy controls are colored in green. The gray area represents the reference range of the panel. *P*-values between conditions are calculated by using the two-sided *t*-test. **c** Heat map of DEGs with functional enrichment in B cells. Gene expression on the sample level is measured by the counts per million mapped reads (CPM), which is then scaled across samples. **d** Percentage of *IGHA* and *IGHG* in BCRs across conditions recovered by scBCR-seq (KD patients $n = 7$, healthy controls $n = 3$). The samples are in the same colors as (**a**). *P*-values are calculated by using the two-sided *t*-test. In (**a**) and (**d**), data are represented as boxplots where the middle line is the median, the lower and upper hinges correspond to the first and third quartiles, and the whiskers extend from the hinge to the farthest data point within a maximum of 1.5× interquartile range. **e** BCR clonotype tracking in each patient. The clonotype composition is represented by stacked bar plots, which are colored according to *IGHV* families. Only clonotypes with clonal size ≥3 are plotted, and proportion of BCRs in these clonotypes are compared before and after therapy by using the two-sided Fisher's exact test. Individual clonotypes with FDR < 0.01 are marked with stars. Source data are provided in the Source Data file.

receptor involved in IL10-mediated anti-inflammatory functions. The discordant interferon responses were also observed in NK cells (Supplementary Figure 20). Similar to B cells, the expression of S100 genes in all these lymphocyte populations were higher in KD patients than those in healthy controls (Fig. 5c). In addition, the gene set enrichment analysis (GSEA)[41] identified significantly coordinated changes of several hallmark gene sets in both CD4+ and CD8+ T cells (FDR < 0.05, Fig. 5d), where oxidative phosphorylation and MYC targets were strongly activated in pre-treatment patients relative to post-treatment patients and healthy controls. It was reported that activated T cells require a substantial increase in energy production including oxidative phosphorylation[42], and MYC is essential in the metabolic reprogramming[43]. In both CD4+ and CD8+ T cells, we observed that a lot of MYC target genes including *MYC* itself were significantly upregulated in pre-treatment KD patients (Fig. 5c, e). Another interesting DEG specific to CD4+ T cells was *FOXP3*, a canonical marker of regulatory T cells. Although we did not found significant change in the percentage of regulatory T cells, the expression of *FOXP3* was upregulated after therapy as described in a previous study[44] (Fig. 5e).

Finally, we tracked the TCR clonotypes based on the scTCR-seq data. T cells with clonal TCRs (clonal size ≥3) were enriched in effector memory CD8+ T cells with high expression of *GZMB* (Supplementary Fig. 21), indicating strong cytolytic activity of the cells. Similar to the dynamics of BCR repertoires, we observed that the percentages of clonal TCRs were obviously increased in most of the patients after IVIG therapy (Fig. 5f), and the increase in the Gini coefficient was also significant ($P = 0.04$, two-sided

*t*-test, Supplementary Fig. 22). We did not find any predominant *TRBV* families in the clonal TCRs (Fig. 5f), and no skewed usage of *TRBV* and *TRAV* could be detected by comparing KD patients and healthy controls (Supplementary Figs. 23 and 24). Again, these results supported that conventional antigens[45–47] rather than a superantigen[48–51] could be responsible for the pathogenesis of KD.

## Discussion
Although great efforts have been made in the research and treatment of KD, many essential questions remain to be clarified, such as what trigger the disease, how the CAL is developed and the mechanism of IVIG therapy. The lack of knowledge has impeded a precise diagnostic test, a better management of IVIG-resistant patients and long-term cardiac sequelae. An overt immune reaction is the most important clinical features of acute KD, which has helped to elucidate etiology and pathogenesis of the disease[10,12]. However, many immunological studies reported conflicting results[15], in part due to the limited resolution of low-throughput screening assays. Bulk transcriptome analyses were performed for peripheral blood, but the signals were overwhelmed by the activation of innate immune system, probably due to the dominate abundance of neutrophils[17,18]. Although the dysregulations of innate immune system were well established in these studies, the roles of adaptive immune system were largely neglected because of the less abundance and high heterogeneity of lymphocytes. In this study, we used scRNA-seq to dissect the complex immune responses of PBMCs in acute KD. We observed remarkable temporal changes in the cell abundance before and

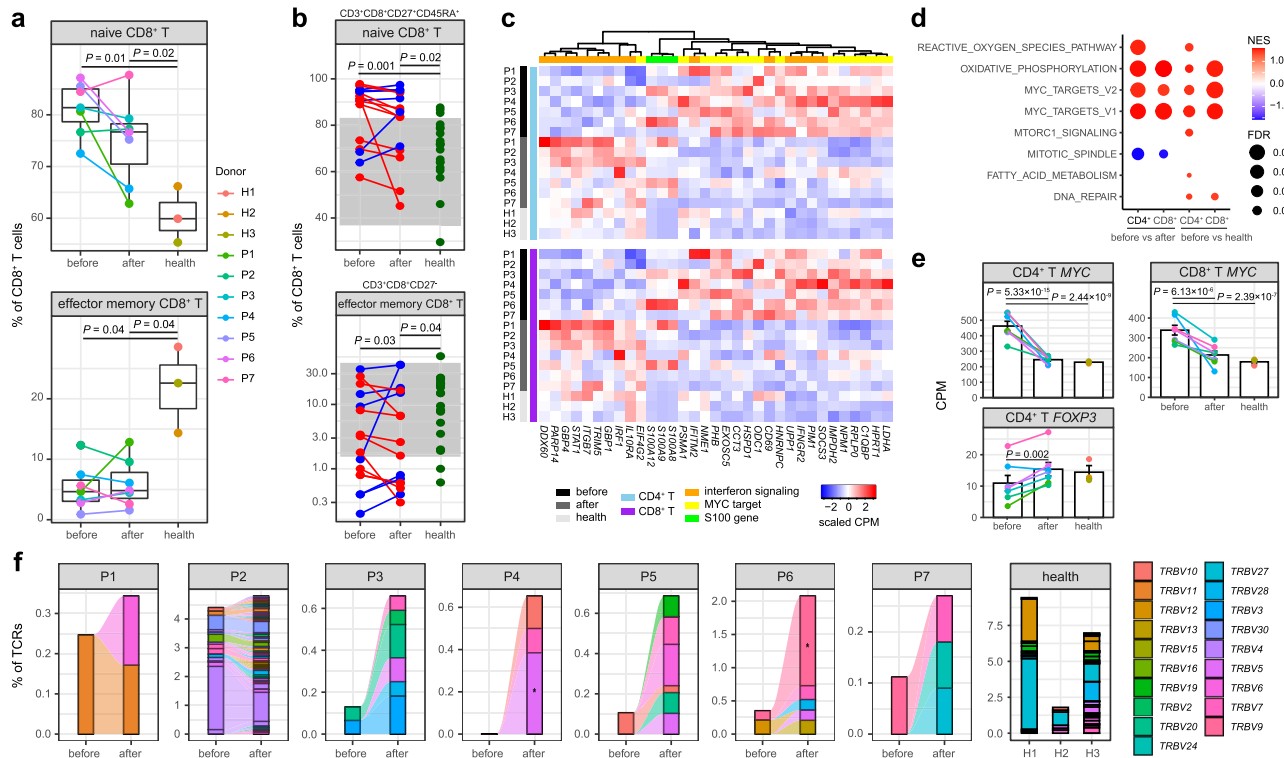

**Fig. 5 Characterization of T cells and TCRs. a** Percentage of naive and effector memory CD8+ T cells across conditions recovered by scRNA-seq (KD patients n = 7, healthy controls n = 3). The middle line of the boxplot is the median, the lower and upper hinges correspond to the first and third quartiles, and the whiskers extend from the hinge to the farthest data point within a maximum of 1.5× interquartile range. The samples are colored according to donors. P-values are calculated by using the two-sided t-test. **b** Percentage of naive (CD3+ CD8+ CD27+ CD45RA+) and effector memory (CD3+ CD8+ CD27−) CD8+ T cells by flow cytometric validation on additional KD patients (n = 16) and healthy controls (n = 20). Patients with a decreased percentage after therapy are colored in red, and patients with an increased percentage after therapy are in blue. Healthy controls are colored in green. The gray areas represent the reference ranges of the panel. P-values between conditions are calculated by using the two-sided t-test. The percentage of effector memory CD8+ T cells is plotted and tested on the log scale. **c** Heat map of DEGs with functional enrichment shared in CD4+ and CD8+ T cells. Gene expression on the sample level is measured by the counts per million mapped reads (CPM), which is then scaled across samples. **d** Significant hallmark gene sets identified by GSEA[41]. The dot size indicates the FDR, and the dot color indicates the normalized enrichment score (NES). A positive NES suggests that the gene set is enriched in upregulated genes before therapy, and a negative NES suggests it is enriched in downregulated genes before therapy. **e** Expression level of MYC and FOXP3 in T cells across conditions (KD patients n = 7, healthy controls n = 3). The bar plot depicts mean CPM and standard error of the mean. The samples are in the same colors as **a**. P-values are calculated with DESeq2[31] (two-sided). **f** TCR clonotype tracking in each patient. The clonotype composition is represented by stacked bar plots, which are colored according to TRBV families. Only clonotypes with clonal size ≥3 are plotted, and proportion of TCRs in these clonotypes are compared before and after therapy by using the two-sided Fisher's exact test. Individual clonotypes with FDR < 0.01 are marked with stars. Source data are provided in the Source Data file.

after IVIG therapy, which were largely consistent with previous studies and our flow cytometric analyses. In particular, the abundance of B cells, CD8+ T cells and NK cells could be helpful in distinguishing KD and other febrile diseases, and predicting the effectiveness of IVIG[22].

We identified unique signatures of gene expression for each cell type. Most of the DEGs arose from monocytes, including many crucial pro-inflammatory mediators and therapeutic targets such as S100 genes[33,34], TNF and IL1B[37]. Most DEGs encoding immunoglobulin receptors showed high expression level before IVIG therapy and returned to normal level after therapy, suggesting that the immunomodulatory effects of IVIG may involve the blockade of activating immunoglobulin receptors[52]. Another intriguing signature was the underexpression of MHC class II genes in acute KD patients relative to healthy controls, which was robustly observed in patients with SARS-CoV-2 infection[53] though the pathological conditions were dissimilar[7–9]. The gene clusters of immunoglobulin receptors and MHC class II receptors were also identified as susceptibility loci of KD in GWAS studies[11], highlighting their essential roles in KD pathogenesis.

Regarding to the lymphocyte populations, we found that although the total B-cell abundance was substantially decreased after therapy, the percentage of plasma cells among the B cells was elevated, which was accompanied by extensive oligoclonal expansion of BCRs and isotype switching to IgG and IgA. We also found significant depletion of CD8+ T cells in PBMCs during acute KD, especially the subset of effector memory cells. In addition, oligoclonal expansion of TCRs was observed after therapy in effector memory CD8+ T cells. All these evidences supported the hypothesis that KD could be caused by an acquired immune response to specific conventional antigens but not superantigens[10,12]. In fact, both oligoclonal IgA plasma cells[54,55] and CD8+ T cells[56] were reported to infiltrate vascular tissue in acute KD, which were proposed as the most important evidence for a respiratory viral etiology[16]. Synthetic antibodies based on the oligoclonal IgAs could be designed to identify potential pathogens of KD[57,58]. However, clinical material available for this kind of study is scarce because accessing the inflamed tissue of KD is impossible in living children. Our results suggested that the oligoclonal BCRs prepared from peripheral blood of patients who

recover from acute KD could show promising opportunity for antigen identification and therapeutic development. Among the CD8[+] T cells, acute KD patients showed a higher percentage of naive cells and lower percentage of effector memory cells than healthy controls, which appeared not affected by IVIG therapy. A possible explanation is that a deficiency in effector memory CD8[+] T cells, which can rapidly respond to antigens, could be a risk factor for KD.

We also identified many biological processes that were rarely investigated in previous studies, such as the promotion of cell cycle progression in B cells, the enhancement of oxidative phosphorylation and the simulation of MYC targets in T cells. Most of the expression patterns were coupled with the over-activated immune responses in acute KD. However, the expression patterns of interferon-induced genes were largely discordant before and after therapy, probably due to both the pro-inflammatory and anti-inflammatory effects of interferon responses[59,60]. Several transcriptome analyses reported a striking absence of interferon-induced gene expression in acute KD, especially compared with virus infection[18,61]. *IFNG* polymorphisms were also associated with the susceptibility of KD and IVIG responsiveness[62]. If KD is triggered by viral infection, the roles of interferon responses in the disease are worth further investigation.

It is notable that MIS-C, a rare and severe hyperinflammatory syndrome temporally associated with SARS-Cov-2 infection, shares some clinical features with KD, including high fever, conjunctivitis, rash, lymphadenopathy, oropharyngeal changes and possible occurrence of coronary artery dilation[7–9]. However, other characteristic findings in MIS-C, such as lymphopenia, shock, cardiac dysfunction and multiple organ failure are rarely seen in KD[7–9]. Moreover, MIS-C affected older children than KD, with the median age up to 9 years old. A recent study has compared the immunopathology of MIS-C, KD and healthy controls[63]. Within the CD4[+] T cells, MIS-C patients had higher abundance of memory CD4[+] T cells and lower abundance of native CD4[+] T cells as compared to KD patients. Consistent with our results, there was no significant difference reported in the CD4[+] T-cell subsets between KD patients and healthy children. Plasma IL17A level was significantly higher in KD patients than that in MIS-C and healthy children, but the expression level of *IL17A* in PBMCs was too low to detect differential expression in our scRNA-seq data (Fig. 3c).

There were several shortcomings in our study. The sample size for scRNA-seq analysis was limited, which affected the statistical power in differential abundance and differential expression analysis. Because all patients diagnosed with acute KD would be treated with IVIG, the intrinsic immune responses to potential pathogens were mixed with the immunomodulatory effects of IVIG. For example, further efforts should be made to distinguish clonally expanded BCRs specific to unknown pathogens and exogenous immunoglobulins after therapy. This study was performed on PBMCs and might not reflect the local inflammatory responses developing in the coronary artery. Also, more patients with various clinical presentations are needed to determine the relationship between immune responses of different cell types and disease outcomes.

## Methods

**Patients.** All donors were recruited from Shanghai Children's Hospital between December 2019 and December 2020. The study was reviewed and approved by the Ethics Committee of Shanghai Children's Hospital (IRB Protocol Number: 2019R081). Informed consent was obtained from the donors and their guardians. The diagnosis of KD was made by using the criteria proposed by the American Heart Association[3], including fever of unknown etiology lasting for ≥5 days, and the presence of five associated symptoms (conjunctivitis, oral changes, extremity changes, rash, cervical lymphadenopathy). Complete KD was diagnosed if at least

four of the symptoms were fulfilled, otherwise incomplete KD. The patients received high-dose IVIG (1 g/kg per day) for two consecutive days combined with oral aspirin (30 mg/kg per day) after diagnosis with KD. CALs were regularly monitored by two-dimensional echocardiography. The seven KD patients subjected to scRNA-seq were aged 1.6–5.4 years, and most of them except P3 were diagnosed with complete KD. The 16 additional KD patients subjected to flow cytometric validation were aged 0.4–7.2 years, and all of them were diagnosed with complete KD. Both groups of patients followed the same timepoints for blood sampling, i.e., the 5th day after the onset of fever before IVIG therapy and 24 h after IVIG therapy and subsidence of fever. All of the patients underwent routine blood tests before and after therapy. They all responded to IVIG therapy and did not develop CALs. Three healthy donors for scRNA-seq and 20 healthy donors for flow cytometric validation with similar ages (1.2–5.5 years) were recruited at routine physical examinations, who presented no recent history of fever, infection or immunization.

**Single-cell preparation and sequencing.** For each donor subjected to scRNA-seq, 2 ml venous blood was collected in EDTA anticoagulant tubes and transferred to the laboratory with ice. The blood was processed within 4 h of collection. PBMCs were isolated by density gradient centrifugation using the Ficoll-Paque medium. The cell viability should exceed 90% determined with trypan blue staining. An appropriate volume of cell suspension was calculated to contain ~12,000 cells for each sample. Single-cell capturing and library construction were performed using the Chromium Next GEM Single Cell V(D)J Reagent Kits v1.1 (10x Genomics) according to the manufacturer's instructions. Briefly, the cell suspension, barcoded gel beads and partitioning oil were loaded onto the 10x Genomics Chromium Chip to generate single-cell Gel Beads-in-Emulsion (GEMs). Captured cells were lysed and the transcripts were barcoded through reverse transcription inside individual GEMs. Then cDNA along with cell barcodes were PCR-amplified. The scRNA-seq libraries were constructed by using the 5' Library Kits (PN-1000165, PN-1000020), and the scBCR-seq and scTCR-seq libraries were constructed by using the V(D)J Enrichment Kits, Human B Cell and T Cell (PN-1000016, PN-1000005). Each sample was processed independently and no cell hashing was applied. The constructed libraries were sequenced on an Illumina NovaSeq platform to generate 2× 150-bp paired-end reads.

**scRNA-seq data analysis.** The raw sequencing data were processed by the Cell Ranger pipeline (v3.0.1, 10x Genomics), including demultiplexing, genome alignment (GRCh38), barcode counting and unique molecular identifier (UMI) counting. The gene-barcode matrix of UMI counts was then analyzed with Seurat (v3.0.2)[20] for quality control, normalization, dimensional reduction, batch effect removal, clustering and visualization. For most samples, the following criteria were applied for quality control: total UMI count between 2,000 and 60,000, and mitochondrial gene percentage <5%. For P1 before therapy, a lower cutoff of total UMI count (1000) was used due to its lower median UMI count per cell. The count matrix was log-normalized, and the top 2,000 most variable genes were identified for dimensional reduction. The samples collected in the first phase of the study (P1–P4 and H1–H3) were integrated to remove batch effects with the canonical correlation analysis (CCA), which identified cross-sample pairs of cells in a matched biological state as anchors and then used the anchors to correct technical differences between samples. The integrated matrix was then scaled, and the top 30 dimensions resulted from the principal component analysis (PCA) were used for the uniform manifold approximation and projection (UMAP). The top PCs were chosen to account for ~95% of the total variance, and the curve flattens out in the scree plot (Supplementary Fig. 25). Meanwhile, the shared nearest neighbor graph-based clustering was performed on the PCA-reduced data to identify cell clusters. The resolution was set to 0.1 to obtain major cell types of PBMCs, and 1.2 to obtain subsets of each cell type. Then the integrated dataset was adopted to annotate new samples collected in the subsequent phase of the study (P5–P7) with Seurat's data transfer utility. The cell identities were determined with SingleR (v1.0.6)[21], which compared the transcriptome of each cell cluster to various reference datasets (human primary cell atlas, Blueprint/ENCODE, Database of Immune Cell Expression, Novershtern hematopoietic data and Monaco immune data). Because inconsistency and ambiguity remained with the automatic assignment, we then refined the cell cluster annotations based on the expression of canonical marker genes.

**Differential expression and functional enrichment analysis.** We performed differential expression analysis for each cell type on the sample level following the recommendation of Bioconductor[64]. A pseudo-bulk expression profile was generated by summing UMI counts together for all cells with the same combination of cell type and sample. Then the differential expression analysis was conducted between conditions by using DESeq2 (v1.28.1)[31], which estimated variance-mean dependence in count data and tested for differential expression based on the negative binomial distribution. We used clusterProfiler (v3.16.0)[65] for function over-representation analyses of DEGs with FDR < 0.05. GO,KEGG[66] and hallmark gene sets in MSigDB (v7.1)[67] served as the gene function databases. GSEA (v4.0.3)[41] was performed based on the counts per million mapped reads (CPM) matrix between conditions, and the hallmark gene sets were used as the database. The permutation type was set to phenotype and the number of permutations was

1000. *P*-values of both over-representation analyses and GSEA were adjusted to FDRs and a gene set was considered significant if FDR < 0.05.

**Comparison with bulk gene expression dataset**. A publicly available cDNA microarray dataset on PBMCs from 19 KD patients[23] was fetched from the GEO database (GSE73577). The fold change of gene expressions before and after IVIG therapy was measured with the microarray, and the differential expression was detected with the two-sided *t*-test. We also transformed our scRNA-seq dataset to a pseudo-bulk expression profile by summing up UMI counts for all PBMCs from a sample. DESeq2[31] was used to calculate the fold change before and after therapy based on the pseudo-bulk expression profile.

**BCR and TCR data analysis**. The scBCR-seq and scTCR-seq data were assembled by the Cell Ranger VDJ pipeline (v3.0.1, 10x Genomics), and reference sequences in IMGT[68] were fetched for annotation. Only cells with productive and paired chains (IGH and IGL/IGK for BCRs, TRA and TRB for TCRs) were preserved. For cells with more than one consensus sequence detected for the same chain type, the one with higher UMI counts was chosen. Clonotypes represented by CDR3 sequences and V gene usages were explored by immunarch (v0.6.5)[69]. The whole repertoire clonality of a sample was measured by the Gini coefficient, where zero expresses perfect equality and one expresses maximal inequality among clonotypes. The proportion of individual clonotypes (size ≥ 3 in at least one sample) was compared between pre- and post-treatment samples with the Fisher's exact test. Clonotypes and expression profiles were matched based on the same cell barcodes.

**Flow cytometric analysis**. We followed a flow cytometric multicolor protocol designed for children to analyze peripheral blood lymphocytes[70]. EDTA-anticoagulated whole blood was transferred to the laboratory and processed immediately after collection. The blood sample was split into three different panels to identify lymphocyte subsets: 1) major lymphocyte populations defined with the BD Multitest IMK Kit (Cat# 662965, 1:2 dilution), including CD19+ B, CD3+ CD4+ T, CD3+ CD8+ T and CD3−CD16+/CD56+ NK cells; 2) B-cell subset panel in which plasma cells are defined with CD19-APC-H7 (Cat# 641395, 1:2 dilution), CD24-PE (Cat# 555428, 1:3 dilution) and CD38-FITC (Cat# 340909, 1:3 dilution); 3) T-cell subset panel in which CD8+ T-cell subsets are defined with CD3-V450 (Cat# 560365, 1:2 dilution), CD8-PerCP (Cat# 652829, 1:5 dilution), CD4-APC-H7 (Cat# 641398, 1:2 dilution), CD27-APC (Cat# 558664, 1:10 dilution) and CD45RA-PE-Cy7 (Cat# 560675, 1:2 dilution). All of the antibodies were purchased from BD Biosciences. After 15 min incubation for staining in the dark at room temperature, red blood cells were lysed with 1 ml RBC lysis buffer (Cat# 555899, BD Biosciences) for 10 min. Then the cells were washed with 1 ml PBS for three times and loaded on BD FACSCanto II. About $1 \times 10^4$ cells were collected in the flow cytometer. The cell populations were gated by using the BD FACSDiva software.

**Reporting summary**. Further information on research design is available in the Nature Research Reporting Summary linked to this article.

## Data availability

The processed sequencing data generated in this study have been deposited in the Gene Expression Omnibus (GEO) database under accession code GSE168732. The raw sequence data generated in this study have been deposited in the National Omics Data Encyclopedia database of Bio-Med Big Data Center, Shanghai Institute of Nutrition and Health, Chinese Academy of Sciences under accession code OEP001162. The raw sequence data are available under restricted access because of data privacy laws, and access can be obtained by reasonable request to the corresponding authors. The bulk gene expression data used in this study are available in the GEO database under accession code GSE73577. Source data are provided with this paper.

## Code availability

Source code used for analyzing the data are available at a public GitHub repository: https://github.com/zhenwang100/scKD [71].

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

## Acknowledgements

This work was supported by the National Key R&D Program of China (2016YFC0901704, 2017YFA0505500), the National Natural Science Foundation of China (32070570), the Strategic Priority Research Program of Chinese Academy of Sciences (XDB38050200), the Science and Technology Service Network Initiative of Chinese Academy of Sciences (KFJ-STS-QYZD-126), the Youth Innovation Promotion Association CAS (2017325), the Shanghai Science and Technology Committee research funding (17411954300, 18411965800, 19495810400), the Shanghai Municipality Health Commission research funding (2019SY025, 2018ZHYL0223), the Shanghai Children's Hospital research funding (2019YLYZ01), the Shanghai Jiaotong University research funding (ZH2018ZDA26), and the Shanghai Hospital Development Center research funding (SHDC12016119). We would like to thank all the donors who contributed samples, Anhui Engineering Laboratory for Big Data of Precision Medicine who contributed computational resource, and Genergy Bio-technology, Co. Ltd who provided scRNA-seq support for this work.

## Author contributions

Z.W., L.X., G.D., Y.L., and M.H. designed research. L.X., S.S., L.C., T.X., Y.H., and M.H. collected samples and clinical information. G.D., G.L., and J.L. performed sequencing. Z.W., G.D., and Y.L. analyzed data. M.X., D.H., Y.Z., and H.Z. performed flow cytometric analyses. Z.W., L.X., Y.L., and M.H. wrote the manuscript.

## Competing interests

The authors declare no competing interests.
