## [Peer Review File · Nature Communications]

Single-cell RNA sequencing of peripheral blood mononuclear cells from acute Kawasaki disease patientsREVIEWER COMMENTS

Reviewer #1 (Remarks to the Author):

In this manuscript, to better characterize the immune response and the transcriptional changes at the single-cell level during Kawasaki Disease (KD), Wang et al. perform single-cell RNA sequencing (ScRNA-seq) analysis of peripheral blood mononuclear cells (PBMCs) isolated from 4 KD patients at 2 different timepoints: during the acute phase and in the convalescent phase (24 hours after IVIG treatment). Similar analysis is performed on PBMCs collected from 3 healthy controls. Based on ScRNA-seq and a flow cytometric analysis of another dataset of KD and IVIG treated patients, the authors show that transcripts of B cells are increased during KD acute phase and significantly decreased in the KD convalescent patients and healthy controls. On the other hand, transcripts related to CD8 T cells and NK cells (although not significant) are decreased during acute KD and this result seems to be confirmed by flow cytometry. By performing BCR and TCR sequencing, the authors show that antibodies are involved in the convalescence phase and that there is no TCR clonal expansion indicating further that KD is not triggered by a superantigen. Although, the participation of the described cellular subsets to KD development has been already previously suggested, this is probably the first study attempting to characterize by ScRNAseq the immune cells involved in KD and during KD convalescent phase. However, this study is largely “descriptive” and has limitations, such as the small number of KD patients involved in the ScRNA-seq study and variability among the results, the lack of depth of the ScRNA-seq analysis, and the fact that the observed changes with the ScRNA-seq are not confirmed at the cellular level on the same patients by the flow cytometry analysis.

Major comments:

- A recurrent issue is the “lack of details” in both the manuscript and the figure legends, which results in confusion when trying to understand the data presented. There is no clear description of the groups analyzed in each specific panels. For Figure 1, it is unclear on which group the UMAP and the dot plot were generated on. Is it a pool of all the KD acute, the KD convalescent or the healthy controls?
- There are some questions regarding how the ScRNA-seq analysis was performed as a clear description is not provided and details on how the data were processed are lacking. There is only 7 lines in the methods (381 to 386) to describe this process and there is no clear description of the packages used or mention of the quality controls performed as well as how the cells were filtered. How were the PCs selected?
- It also seems that the cell clusters annotation was performed “manually” based on canonical markers. There are today multiple new tools that would allow to annotate the immune cells clusters (SingleR; SCSA, scCATCH). The authors should use one of those to at least confirm their cell clusters annotation and may even end up with a more refined annotation.
- It would have been more beneficial to present the UMAPS separated by groups, such KD acute, KD convalescent and healthy controls. Performing comparative analysis where the different conditions side-by side can be observed should give a global picture of the changes in immune cells during disease and convalescence as compared to healthy controls. Similarly, it is unknown if panels A of Figures 1, 3, 4, 5 refer to data generated from acute KD only Showing the proportion of those cells in all the groups, from acute KD, control and convalescent KD patients (this can be done with Seurat)
- In figure 2, the authors compared the Sc-RNAseq results with a flow cytometric analysis performed on a “dataset” of n=125 acute KD patients before and after IVIG therapy. However, the description of this dataset is completely missing in the manuscript. How were the patients selected/recruited, how was the blood processed, which markers were used for the flow cytometric analysis, was the IVIG treated timepoint the same than the one used for the scRNAseq analysis (24 hours?).

- Are the 4 KD patients before and after IVIG treatment used for the Sc-RNA seq analysis included in the flow cytometric dataset as well? If yes, it might be worth to look if the changes observed at the transcriptomic level by the ScRNA-seq analysis are also present in the flow cytometric analysis.
- Why did the authors choose an FDR p-value of 0.1 and not 0.05? If you look at figure S4, a more stringent p-value would decrease by 2 fold the numbers of DEGs and may be more specific for the subsequent pathways analysis.
- The authors show that monocytes related transcripts are upregulated during acute KD and decreased after IVIG treatment (Figure 3A, B- unsure if B refer to the ScRNAseq or the flow cytometry analysis). This is not surprising as the contribution of monocytes and innate cells (neutrophils) to the KD inflammatory reaction is well described and established. However, the role of IL-1b, as well as TNF-a and IL-6 in KD development is also clearly established. In fact, Anakinra, an IL-1 receptor antagonist, is used in clinical trials to treat IVIG resistant patients (see ref. Kone-Paut I et al. Arthritis Rheumatol. 2020 Aug 11., Koné-Paut, I. (2018). Autoimmun Rev 17(8), 768 – 774). And IL-1b production by monocytes appears to be key in this process (Armaroli, G. et al. (2019). Arthritis Rheumatol 71(5), 792 - 804.) It is highly surprising that the authors do not “pick up” in their all analysis any IL-1b, TNF-a or IL-6 related genes or pathways signatures. It is also surprising that neutrophils are not identified as a cluster in the ScRNAseq analysis. Could the authors explain why and discuss those results?
- Figure 4, panel E would benefit to include the health controls in the analysis.
- Could the author discuss more why IVIG treatment does not affect the CD8 cells compartment?
- Lines 286 to 288 of the discussion: The authors discuss previous bulk-RNA seq analysis on PBMCs from KD and febrile controls and mention that this analysis might have been overwhelmed by a neutrophil “signature”. How did they reach such conclusion? It is also surprising, given the established role of neutrophils in KD development, that this population is not “detected” by the scRNAseq analysis. Could the authors explain and develop that part?
- The figure legends lacks the minimum details needed to understand the different panels.

Minor comments:

- The authors consider that MIS-C or PIM-TS which develops in children is similar to Kawasaki disease. However, it is clearly established now that MIS-C and Kawasaki Disease are two different entities and MIS-C is more related to toxic shock syndrome (see the following references: Rowley, A. (2020). Nature Reviews Immunology, Whittaker, E. et al. (2020) JAMA 324(3) , Cheung, E. et al. (2020) JAMA 324(3), Rowley, A. et al. (2020). Journal of Clinical Investigation (2020)). Therefore, the following lines (74 to 77 and 302 to 304) should be revised.
- There is no IRB # associated to this study
- Single-cell sequencing in all the manuscript should be changed by single-cell RNA sequencing

Reviewer #2 (Remarks to the Author):

1. The patients received high-dose IVIG (1 g/kg per day) this is not the standard treatment for KD. why use this protocol ?
2. Author should mentioned about dosage of aspirin.
3. IRB should be included, blood amount and patient inform consent should be mentioned.

4. What is the major finding of this study?
5. about FCGR2A and FCGR2B should be discussed and cite references.

Reviewer #3 (Remarks to the Author):

In this manuscript, Wang and colleagues utilized single cell RNA sequencing to study Kawasaki disease (KD). KD is a form of systemic vasculitis that primarily affect young children and up to 30% of patients can develop coronary aneurysms. The pathophysiology of KD is not well understood. Although intravenous immunoglobulins (IVIG) is effective in preventing the development of coronary aneurysms, the mechanism by which IVIG mediates the therapeutic effect is also unclear. There is a clear need to better understand the immunologic underpinning of KD. The authors performed scRNAseq on peripheral blood mononuclear cells from 4 patients with KD, before and after IVIG treatment, and also from 3 healthy controls. The performed a range of analysis including cell count, differential gene expression, gene set enrichment analysis, as well as BCR and TCR sequencing. While a recent study assessed scRNAseq of monocytes in KD (Syrimi et al, MedRxiv 2020), this study represent a more comprehensive analysis of various leukocyte subsets. However, the number of subjects is rather small without any approach to data validation. In the current form, the study is mostly descriptive and it is unclear how the findings help advance our understanding of KD.

Major comments:

1) The first part of the study is focused on comparing cell populations in the KD samples pre/post IVIG and also with healthy controls. These data seem to be more supplemental as much of the findings were in agreement with existing literature and can be derived readily using flow cytometry as the authors indicated.

2) The findings are in part dependent on cell input and the method section is not clear on how many cells from each sample were incorporated in each run. The authors should indicate if hashing applied and how batch variations were addressed.

3) While some of the gene expression findings were of interest, the pathways involved are broad and there was no attempt to validate the data using another method (qPCR, flow cytometry etc). The decreased IFN signature in pre-treatment KD samples compared to post-treatment and healthy controls is also counterintuitive compared to what is known in the literature. Validation studies are necessary to truly demonstrate how the current data can overcome limitations of previous studies.

4) Similarly, there was no attempt to verify the potentially interesting BCR / TCR data. Without validation studies or further exploration of the specificity of these receptors, inference such as "suggesting that antibody-mediated responses might be involved in the recovery of acute KD" and "the antibodies might also be identified from peripheral blood of patients recovered from acute KD" are purely speculative and should be part of the Discussion (instead of Results).

5) The number of patients (n=4) is fairly small to capture the heterogeneity of KD. One patient had incomplete KD. A key feature of KD to address, both in terms of diagnosis and treatment, is the development of coronary aneurysms. None of the 4 patients in the study had coronary aneurysms, which may reflect timely treatment. It be helpful if the authors expand on how their findings help advance our understanding of KD.

Minor comments

Page 7 Line 161 – MDCs are not monocytes and should not be classified as such.

Responses to Reviewer #1:

In this manuscript, to better characterize the immune response and the transcriptional changes at the single-cell level during Kawasaki Disease (KD), Wang et al. perform single-cell RNA sequencing (ScRNA-seq) analysis of peripheral blood mononuclear cells (PBMCs) isolated from 4 KD patients at 2 different timepoints: during the acute phase and in the convalescent phase (24 hours after IVIG treatment). Similar analysis is performed on PBMCs collected from 3 healthy controls. Based on ScRNA-seq and a flow cytometric analysis of another dataset of KD and IVIG treated patients, the authors show that transcripts of B cells are increased during KD acute phase and significantly decreased in the KD convalescent patients and healthy controls. On the other hand, transcripts related to CD8 T cells and NK cells (although not significant) are decreased during acute KD and this result seems to be confirmed by flow cytometry. By performing BCR and TCR sequencing, the authors show that antibodies are involved in the convalescence phase and that there is no TCR clonal expansion indicating further that KD is not triggered by a superantigen. Although, the participation of the described cellular subsets to KD development has been already previously suggested, this is probably the first study attempting to characterize by ScRNAseq the immune cells involved in KD and during KD convalescent phase. However, this study is largely “descriptive” and has limitations, such as the small number of KD patients involved in the ScRNA-seq study and variability among the results, the lack of depth of the ScRNA-seq analysis, and the fact that the observed changes with the ScRNA-seq are not confirmed at the cellular level on the same patients by the flow cytometry analysis.

We thank the reviewer’s constructive comments to our manuscript and we have revised the manuscript taking all of the suggestions into account. Particularly, we recruited 3 more KDs for the scRNA-seq analysis before and after IVIG therapy (now a total of 7), and performed flow cytometric validation on additional 16 KDs at the same time point as scRNA-seq. Please find our point-to-point responses to the comments below.

Major comments:

- A recurrent issue is the “lack of details” in both the manuscript and the figure legends, which results in confusion when trying to understand the data presented. There is no clear description of the groups analyzed in each specific panels. For Figure 1, it is unclear on which group the UMAP and the dot plot were generated on. Is it a pool of all the KD acute, the KD convalescent or the healthy controls?

We provided more detailed description of the methods and figure legends. The original Figure 1 was a pool of cells from all samples. In the revised manuscript, we presented the UMAP plot by different conditions (before therapy, after therapy and healthy controls) as the reviewer suggested.

- There are some questions regarding how the ScRNA-seq analysis was performed as a clear description is not provided and details on how the data were processed are lacking. There is only 7 lines in the methods (381 to 386) to describe this process and there is no clear description of the packages used or mention of the quality controls performed as well as how the cells were filtered. How were the PCs selected?

We detailed the scRNA-seq data analysis in Methods (Line 447-476). Briefly, the Seurat workflow was used for quality control, normalization, dimensional reduction, batch effect removal, clustering and visualization. The cells were filtered with UMI count between 2,000 and 60,000, and mitochondrial gene percentage < 5%. The top 30 PCs were chosen to account for ~95% of the total variance, and the curve flattens out in the PCA scree plot (Supplementary Figure S24).

- It also seems that the cell clusters annotation was performed “manually” based on canonical markers. There are today multiple new tools that would allow to annotate the immune cells clusters (SingleR; SCSA, scCATCH). The authors should use one of those to at least confirm their cell clusters annotation and may even end up with a more refined annotation.

We used SingleR for automatic cell type annotations based on five recommended reference immune datasets (Line 470-476, Supplementary Fig. S1). SingleR performed well on major cell compartments (monocytes, B, T, NK), but ambiguity and inconsistency existed by using different reference datasets. Therefore, we checked canonical markers to refine the cell cluster annotations. Combining SingleR and canonical markers, we were able to dissect several cell subsets that were missing in our original manuscript, such as $\gamma\delta$ T cells and HSPCs (Fig. 1).

- It would have been more beneficial to present the UMAPS separated by groups, such KD acute, KD convalescent and healthy controls. Performing comparative analysis where the different conditions side-by-side can be observed should give a global picture or the changes in immune cells during disease and convalescence as compared to healthy controls. Similarly, it is unknown if panels A of Figures 1, 3, 4, 5 refer to data generated from acute KD only Showing the proportion of those cells in all the groups, from acute KD, control and convalescent KD patients (this can be done with Seurat)

Good suggestion. We now presented the UMAP plot by different conditions and showed the percentage of each cell subset across the conditions. Fig. 1 gave a global picture of all PBMCs, and Supplementary Fig. S5, S8 and S16 gave the proportion among each compartment.

- In figure 2, the authors compared the Sc-RNAseq results with a flow cytometric analysis performed on a “dataset” of n=125 acute KD patients before and after IVIG therapy. However, the description of this dataset is completely missing in the manuscript. How were the patients selected/recruited, how was the blood processed, which markers were used for the flow cytometric analysis, was the IVIG treated timepoint the same than the one used for the scRNAseq analysis (24 hours?).

The dataset of 125 KDs was part of our previous studies (Chen L, et al. Front Pediatr. 2020;8:462367). However, we realized that the dataset was not quite matched to our current scRNA-seq design because the 125 patients were recruited at various time point and no post-treatment samples were collected. So, during the revision of the manuscript, we recruited additional 16 complete KDs for flow cytometric validation (described at Line 419-424). These patients were sampled at the same time point as those for scRNA-seq (the 5th days after the onset

of fever before IVIG therapy and 24 hours after IVIG therapy and subsidence of fever). We adopted a flow cytometric multicolor protocol designed for children to analyze peripheral blood lymphocytes (Ding Y, et al. J Allergy Clin Immunol. 2018; 142:970). We detailed the protocol including blood processing and flow cytometry markers in Methods (Line 505-522).

- Are the 4 KD patients before and after IVIG treatment used for the Sc-RNA seq analysis included in the flow cytometric dataset as well? If yes, it might be worth to look if the changes observed at the transcriptomic level by the ScRNA-seq analysis are also present in the flow cytometric analysis.

Because the blood from each child approved for the study is quite limited, usually we did not have enough materials to perform both scRNA-seq and flow cytometric validation on the same child. So we have to recruit additional patients for flow cytometric analysis. Anyway, the change in cell percentage observed in scRNA-seq can also be confirmed by flow cytometry on the biological replicates (Fig.2A-B, Fig. 4 A-B, Fig. 5A-B).

- Why did the authors choose an FDR p-value of 0.1 and not 0.05? If you look at figure S4, a more stringent p-value would decrease by 2 fold the numbers of DEGs and may be more specific for the subsequent pathways analysis.

We chose a FDR of 0.1 because of the small sample size in our original manuscript. In the revised version, we increased the sample size for scRNA-seq (7 pairs), and changed to a FDR of 0.05 for both differential expression and functional enrichment analysis. The stringent cutoff resulted in more specific pathways, especially for GSEA (Fig. 5D).

- The authors show that monocytes related transcripts are upregulated during acute KD and decreased after IVIG treatment (Figure 3A, B- unsure if B refer to the ScRNaseq or the flow cytometry analysis). This is not surprising as the contribution of monocytes and innate cells (neutrophils) to the KD inflammatory reaction is well described and established. However, the role of IL-1b, as well as TNF-a and IL-6 in KD development is also clearly established. In fact, Anakinra, an IL-1 receptor antagonist, is used in clinical trials to treat IVIG resistant patients (see ref. Kone-Paut I et al. Arthritis Rheumatol. 2020 Aug 11., Koné-Paut, I. (2018). Autoimmun Rev 17(8), 768 – 774). And IL-1b production by monocytes appears to be key in this process (Armaroli, G. et al. (2019). Arthritis Rheumatol 71(5), 792 - 804.) It is highly surprising that the authors do not “pick up” in their all analysis any IL-1b, TNF-a or IL-6 related genes or pathways signatures. It is also surprising that neutrophils are not identified as a cluster in the ScRNaseq analysis. Could the authors explain why and discuss those results?

We realized the importance of the cytokines and inspected the expression of these cytokine genes in our scRNA-seq data. For IL1B, it indeed showed decreased expression after therapy, but only significant with nominal p-value ($P = 4.91 \times 10^{-3}$) but not corrected p-value (FDR = 0.06, Fig. 3D) if the cutoff is stringent. For TNF, the decreased expression was significant with corrected p-value (FDR = 0.001, Fig. 3D). IL6, however, could not be picked up due to its low expression in PBMCs. To give the readers a more global picture, we listed the expression of all cytokine genes

possibly associated with KDs across PBMCs (Fig. 3C) and marked those with nominal p-value < 0.05, which was informative for candidate gene study.

We explained the issue of neutrophils to the following comments.

- Figure 4, panel E would benefit to include the health controls in the analysis.

We included the clonotypes of healthy controls in both BCR and TCR analysis (Fig. 4E and Fig. 5F). In addition, to compare the clonotypes more quantitatively, we calculated the Gini coefficient for pre-treatment, post-treatment and healthy donors (Supplementary Fig. S11 and S21).

- Could the author discuss more why IVIG treatment does not affect the CD8 cells compartment?

KDs showed higher percentage of native CD8 T cells and lower percentage of effector memory CD8 T cells than healthy controls (Fig. 5A). We think a possible explanation is that the percentage of CD8 T cell subsets is not associated with IVIG treatment, but associated with the risk of KD (discussed at Line 377-381).

- Lines 286 to 288 of the discussion: The authors discuss previous bulk-RNA seq analysis on PBMCs from KD and febrile controls and mention that this analysis might have been overwhelmed by a neutrophil "signature". How did they reach such conclusion? It is also surprising, given the established role of neutrophils in KD development, that this population is not "detected" by the scRNAseq analysis. Could the authors explain and develop that part?

In the two bulk-RNA studies on whole blood we cited, Popper et al. (Genome Biol. 2007;8:R261) concluded that "Acute KD is characterized by dynamic and variable gene-expression programs that highlight the importance of neutrophil activation state and apoptosis in KD pathogenesis." And Hoang et al. (Genome Med. 2014;6:541) concluded that "The overwhelming signature for acute KD involved signaling pathways of the innate immune system." In fact, our routine blood tests of the 7 patients for scRNA-seq also showed a substantially high abundance of neutrophils in acute KDs (Supplementary Table S2). Although the bulk-RNA analyses can well reveal the innate immune responses especially neutrophils, expression signatures of less abundant cell types could be missing. We discussed this at Line 335-341.

PBMCs are subsets of peripheral blood cells that only contain lymphocytes and monocytes. Neutrophils are highly abundant in whole blood, but they are usually removed during PBMC preparation. We checked recent publications about scRNA-seq on PBMCs (Zhang et al. Nat Immunol. 2020;21:1107, Wen et al. Cell Discovery 2020;6:31, Hashimoto et al. Proc Natl Acad Sci U S A. 2019;116:24242), and neutrophils can also not be detected in these studies.

- The figure legends lacks the minimum details needed to understand the different panels.

Sorry for the ambiguity. We provided more detailed descriptions of figure legends.

Minor comments:

- The authors consider that MIS-C or PIM-TS which develops in children is similar to Kawasaki disease. However, it is clearly established now that MIS-C and Kawasaki Disease are two different entities and MIS-C is more related to toxic shock syndrome (see the following references: Rowley, A. (2020). Nature Reviews Immunology, Whittaker, E. et al. (2020) JAMA 324(3) , Cheung, E. et al. (2020) JAMA 324(3), Rowley, A. et al. (2020). Journal of Clinical Investigation (2020)). Therefore, the following lines (74 to 77 and 302 to 304) should be revised.

We realized important differences between MIS-C and KD. Line 60-66 and Line 355-357 were revised.

- There is no IRB # associated to this study.

IRB Protocol # 2019R081 was added (Line 409).

- Single-cell sequencing in all the manuscript should be changed by single-cell RNA sequencing

We used single-cell RNA sequencing or scRNA-seq throughout the manuscript.

Reponses to Reviewer #2:

We thank the reviewer's constructive comments to our manuscript and we have revised the manuscript taking all of the suggestions into account. Please find our point-to-point responses below.

1. The patients received high-dose IVIG (1 g/kg per day) this is not the standard treatment for KD. why use this protocol ?

The standard protocol of IVIG treatment is 2 g/kg as a single fusion [1]. However, this protocol usually results in high cost for KD patients, especially for those with high body weight. Recent clinical studies in both China [2] and Japan [3] have shown that using a dosage of 1 g/kg once or for 2 consecutive days are a more cost-efficient therapy option, and no significant difference was found in the outcomes compared with the dosage of 2 g/kg. In this study, we used 1 g/kg for 2 consecutive days.

2. Author should mentioned about dosage of aspirin.

We added the dosage of aspirin (30 mg/kg per day, Line 416), which is a standard moderate dosage [1].

3. IRB should be included, blood amount and patient inform consent should be mentioned.

This study was reviewed and approved by the Ethics Committee of the Shanghai Children's Hospital. We included IRB Protocol # 2019R081 and patient inform consent (Line 408-410). 2 ml venous blood was used for analysis (mentioned at Line 429).

4. What is the major finding of this study?

We summarized the major findings of the study in Discussion. Briefly, 1) we presented a landscape of immune responses of acute KD at single-cell resolution. 2) Monocytes are the major source of pro-inflammatory mediators and therapeutic targets in PBMCs. 3) The involvement of plasma cells and effector memory CD8 T cells, as well as oligoclonal expansion of BCRs and TCRs, provide new evidence that KD can be caused by specific conventional antigens. 4) We identified new biological processes underlying the immune dysregulation of each cell compartment, such as the activation of MYC targets in T cells.

5. about FCGR2A and FCGR2B should be discussed and cite references.

FCGR2A was usually considered as an activating Fc receptor and FCGR2B was usually considered as an inhibitory Fc receptor. However, both FCGR2A and FCGR2B were downregulated after IVIG therapy. These results were consistent with a previous report [4], and were mentioned at Line 194-199.

Reference

- [1] McCrindle et al. Diagnosis, Treatment, and Long-Term Management of Kawasaki Disease: A Scientific Statement for Health Professionals From the American Heart Association. *Circulation*. 2017;135(17):e927-e999. doi: 10.1161/CIR.0000000000000484.
- [2] He et al. Randomized Trial of Different initial IVIG Regimens in Kawasaki Disease. *Pediatr Int*. 2021. doi: 10.1111/ped.14656.
- [3] Suzuki et al. High-dose versus low-dose intravenous immunoglobulin for treatment of children with Kawasaki disease weighing 25 kg or more. *Eur J Pediatr*. 2020;179(12):1901-1907. doi: 10.1007/s00431-020-03794-2.
- [4] Ichiyama et al. Intravenous immunoglobulin does not increase FcγRIIB expression on monocytes/macrophages during acute Kawasaki disease. *Rheumatology (Oxford)*. 2005;44(3):314-7. doi: 10.1093/rheumatology/keh488.

Responses to Reviewer #3:

In this manuscript, Wang and colleagues utilized single cell RNA sequencing to study Kawasaki disease (KD). KD is a form of systemic vasculitis that primarily affect young children and up to 30% of patients can develop coronary aneurysms. The pathophysiology of KD is not well understood. Although intravenous immunoglobulins (IVIG) is effective in preventing the development of coronary aneurysms, the mechanism by which IVIG mediates the therapeutic effect is also unclear. There is a clear need to better understand the immunologic underpinning of KD. The authors performed scRNAseq on peripheral blood mononuclear cells from 4 patients with KD, before and after IVIG treatment, and also from 3 healthy controls. The performed a range of analysis including cell count, differential gene expression, gene set enrichment analysis, as well as BCR and TCR sequencing. While a recent study assessed scRNAseq of monocytes in KD (Syrimi et al, MedRxiv 2020), this study represent a more comprehensive analysis of various leukocyte subsets. However, the number of subjects is rather small without any approach to data validation. In the current form, the study is mostly descriptive and it is unclear how the findings help advance our understanding of KD.

We thank the reviewer's constructive comments to our manuscript and we have revised the manuscript taking all of the suggestions into account. Particularly, we recruited 3 more KDs for the scRNA-seq analysis before and after IVIG therapy (now a total of 7). We also performed flow cytometric validation on additional 16 KDs, especially for our newly identified changes in plasma cells and effector memory CD8 T cells. Please find our point-to-point responses to the comments below.

Major comments:

1) The first part of the study is focused on comparing cell populations in the KD samples pre/post IVIG and also with healthy controls. These data seem to be more supplemental as much of the findings were in agreement with existing literature and can be derived readily using flow cytometry as the authors indicated.

We now improved the resolution of cell populations in the first part, and presented a global picture of changes in cell populations across treatment conditions as suggested by Reviewer #1 (Fig. 1). Although the changes of major cell compartments are in agreement with existing literature and flow cytometry (Fig. 2), we think this part is indispensable to demonstrate the quality of our scRNA-seq data.

2) The findings are in part dependent on cell input and the method section is not clear on how many cells from each sample were incorporated in each run. The authors should indicate if hashing applied and how batch variations were addressed.

We provided a more detailed description of single-cell preparation in Methods (Line 428-446). Briefly, ~12,000 cells were incorporated from each sample (detailed in Supplementary Table S3). Each sample was sequenced independently and no cell hashing was applied. The batch effect was removed in the following bioinformatics analysis by using Seurat (Line 457-461), which is the

most widely used computational tool for batch variation correction.

3) While some of the gene expression findings were of interest, the pathways involved are broad and there was no attempt to validate the data using another method (qPCR, flow cytometry etc). The decreased IFN signature in pre-treatment KD samples compared to post-treatment and healthy controls is also counterintuitive compared to what is known in the literature. Validation studies are necessary to truly demonstrate how the current data can overcome limitations of previous studies.

In the revised manuscript, we validated the changes in plasma cells and effector memory CD8 T cells with flow cytometric analysis on additional 16 KDs (Fig. 4A-B, Fig. 5A-B, Supplementary Fig. S9 and S17). We also used a public bulk-RNA dataset of PBMCs to validate our global expression patterns (Supplementary Fig. S3). We are sorry for the misleading interpretation of interferon signature, which was caused by a loose setting in GSEA. The interferon response genes were enriched, but their expression patterns were quite discordant before and after therapy, especially in T cells (Fig. 5C). So the original result of decreased interferon signature before therapy is not warranted. In the revised manuscript, we chose a more stringent cutoff for pathway analysis as suggested by Reviewer #1, and the obtained pathway activities are more specific (Fig. 5D).

4) Similarly, there was no attempt to verify the potentially interesting BCR / TCR data. Without validation studies or further exploration of the specificity of these receptors, inference such as “suggesting that antibody-mediated responses might be involved in the recovery of acute KD” and “the antibodies might also be identified from peripheral blood of patients recovered from acute KD” are purely speculative and should be part of the Discussion (instead of Results).

In the revised manuscript, we increased the sample size for BCR and TCR analysis ($n = 7$) and confirmed significant oligoclonal expansions of BCR and TCR in most cases (Fig. 4E and Fig. 5F). We agreed that the specificity of these receptors needed further exploration, and moved inference based on BCR and TCR to Discussion following the Reviewer’s suggestion (Line 367-377).

5) The number of patents ($n=4$) is fairly small to capture the heterogeneity of KD. One patient had incomplete KD. A key feature of KD to address, both in terms of diagnosis and treatment, is the development of coronary aneurysms. None of the 4 patents in the study had coronary aneurysms, which may reflect timely treatment. It be helpful if the authors expand on how their findings help advance our understanding of KD.

We increased the patient number to $n = 7$ in the revised manuscript. However, because these patients were randomly recruited, none of them developed CALs after timely IVIG therapy. We recognized the shortcoming of limited heterogeneity in Discussion (Line 402-404). We also revised the discussion section to clarify how the findings may help to advance our understanding of KDs. In summary, 1) we presented a landscape of immune responses of acute KD at single-cell resolution. 2) Monocytes are the major source of pro-inflammatory mediators and therapeutic targets in PBMCs. 3) The involvement of plasma cells and effector memory CD8 T cells, as well

as oligoclonal expansion of BCRs and TCRs, provide new evidence that KD can be caused by specific conventional antigens. 4) We also identified new biological processes worth further investigation.

Minor comments

Page 7 Line 161 – MDCs are not monocytes and should not be classified as such.

We excluded mDCs in the monocyte subsets (Line 173-174).

REVIEWER COMMENTS

Reviewer #1 (Remarks to the Author):

I thank the authors for addressing my previous comments.

1) The small number of patients enrolled in the healthy control group (n=3) scRNA-seq analysis is still a concern. The authors did validate some of the observations obtained by scRNA-seq between KD patients pre and post-IVIG treatment by flow cytometry, the immunological differences reported as significant by the scRNAseq analysis between KD (pre and post-IVIG) vs healthy controls, such as lower CD8 T cells compartment in KD vs Healthy control, were not further validated. Although, the authors did mention that the size of the groups are a limitation of the study.

2) Another limitation that is not discussed is this study is performed on PBMCs and reflects the "systemic immune profiles" and might not be reflective the local inflammatory immune response/infiltrates in the coronary artery.

3) New Figure 1 figure legend, authors should indicate in the figure legend if the UMAP plots are representative of 1 patient or a pool of all the patients in the group.

4) Lines 135 to 142. The authors compare the results of the scRNA-seq dataset with a publicly available PBMCs microarrays dataset on KD before and after IVIG treatment. However, how this analysis was performed does not appear to be described in the methods section.

5) At multiple occurrence, the authors write "KDs" (line 59,157, ..) – I would suggest to replace KDs by "KD patients".

6) Line 105: "All patients were IVIG sensitive responders" should maybe be replaced by "All patients responded to IVIG treatment".

Reviewer #3 (Remarks to the Author):

In this revised manuscript, the authors addressed some of the revision and increased the number of subjects. While the overall analysis is interesting and novel, it remains a concern to me that much of the findings are descriptive without concrete evidence to support insight to the pathophysiology of KD. As a practitioner caring for many patients with KD each year, I do not feel these data have changed how I think about and manage the disease. The number of subjects also should be explicitly stated in the abstract to avoid being vague.

Reviewer #4 (Remarks to the Author):

In this paper, PMBCs were collected from 7 Kawasaki disease (KD) patients before and after IVIG therapy, and 3 age-matched healthy donors. This study profiled single-cell landscape of both innate and adaptive immune responses related to disease and treatment. The study used three technologies, including scRNA-seq, scBCR-seq and scTCR-seq, and described a dynamic immune cell landscape, including decreasing of CD8 T cells in acute KDs, oligoclonal expansions of both B-cell and T-cell after treatment, as well as the immune dysregulation of each cell compartment. All these results provided new insights into KD pathogenesis and potential therapeutic strategies. Thus, the paper could be accepted if the authors could add additional data and modifications.

1, The authors have performed immense of comparisons between cell ratios in KD patients before and after IVIG therapy by pairing statistics, and have described the changes in details. However, they did not execute all corresponding comparisons between health and KD patients, which might be useful for understanding KD pathogenesis.

2, A recent published paper (Consiglio et al., 2020, Cell 183, 968–981) also investigated in the children immunology with COVID-19, KD and health controls. That study also described cell ratio

changes and indicated that IL17A might drive Kawasaki disease. This study would be more convincing by compare and discuss their findings with that paper.

Minor suggestions:

1. The percentage number showed on the UMAP plot (Figure 1) makes it difficult to find specific cluster number on the UMAP plot. Instead, the percentage number could be moved to a supplemental table or figure. Try ratio-of-observe/expectation scores and plot a heatmap figure could simply and directly present the enrichment of cell type in patients and health controls.
2. In Figure 2B, the authors validated the percentage of lymphocyte compartments by FACS experiments on additional 16 KDs samples before and after treatment. Please illustrate the FACS results on normal controls and comparisons between KD and normal controls.

Responses to Reviewer #1:

I thank the authors for addressing my previous comments.

We thank the reviewer for the careful review and valuable suggestions. We have revised the manuscript taking all of the suggestions into account. Please find our point-to-point responses to the comments below.

1) The small number of patients enrolled in the healthy control group (n=3) scRNA-seq analysis is still a concern. The authors did validate some of the observations obtained by scRNA-seq between KD patients pre and post-IVIG treatment by flow cytometry, the immunological differences reported as significant by the scRNA-seq analysis between KD (pre and post-IVIG) vs healthy controls, such as lower CD8 T cells compartment in KD vs Healthy control, were not further validated. Although, the authors did mention that the size of the groups are a limitation of the study.

To address the concern, we performed flow cytometric analysis for additional 20 healthy children. The data are shown on Fig. 2B, Fig. 4B and Fig. 5B. For example, the lower percentage of CD8 T cells in KD vs healthy controls could be validated (Fig. 2B).

2) Another limitation that is not discussed is this study is performed on PBMCs and reflects the “systemic immune profiles” and might not be reflective the local inflammatory immune response/infiltrates in the coronary artery.

We agree with the limitation and added the discussion (Line 423-426).

3) New Figure 1 figure legend, authors should indicate in the figure legend if the UMAP plots are representative of 1 patient or a pool of all the patients in the group.

The UMAP plots are a pool of all patients in each group. We specified this in figure legend (Line 761-762). Please note that to make Figure 1 more easy to read, we only preserved the percentage numbers of major cell compartment in the plot, and moved the percentage number of subtypes to supplementary Fig. S3 (as suggested by Reviewer #4).

4) Lines 135 to 142. The authors compare the results of the scRNA-seq dataset with a publicly available PBMCs microarrays dataset on KD before and after IVIG treatment. However, how this analysis was performed does not appear to be described in the methods section.

We added the method for comparison with the microarray data (Line 517-524). Basically, the scRNA-seq data were transformed to a pseudo-bulk expression profile by summing up UMI counts for all PBMCs from a sample. Then fold changes between the two datasets can be compared.

5) At multiple occurrence, the authors write “KDs” (line 59,157, ..) – I would suggest to replace

KDs by “KD patients”.

We used “KD patients” throughout the manuscript.

6) Line 105: “All patients were IVIG sensitive responders” should maybe be replaced by “All patients responded to IVIG treatment”.

This sentence was revised as suggested (Line 104-105).

Responses to Reviewer #3:

In this revised manuscript, the authors addressed some of the revision and increased the number of subjects. While the overall analysis is interesting and novel, it remains a concern to me that much of the findings are descriptive without concrete evidence to support insight to the pathophysiology of KD. As a practitioner caring for many patients with KD each year, I do not feel these data have changed how I think about and manage the disease. The number of subjects also should be explicitly stated in the abstract to avoid being vague.

We thank the reviewer for the feedback. As an initial study to present a single-cell landscape of immune responses of acute KD, we think our data add novel results to existing literature on KD. In the revised manuscript, we performed flow cytometry analysis of 20 additional healthy controls for further validation. We specified the sample size for scRNA-seq in abstract as suggested (Line 33-34).

Responses to Reviewer #4:

In this paper, PMBCs were collected from 7 Kawasaki disease (KD) patients before and after IVIG therapy, and 3 age-matched healthy donors. This study profiled single-cell landscape of both innate and adaptive immune responses related to disease and treatment. The study used three technologies, including scRNA-seq, scBCR-seq and scTCR-seq, and described a dynamic immune cell landscape, including decreasing of CD8 T cells in acute KDs, oligoclonal expansions of both B-cell and T-cell after treatment, as well as the immune dysregulation of each cell compartment. All these results provided new insights into KD pathogenesis and potential therapeutic strategies. Thus, the paper could be accepted if the authors could add additional data and modifications.

We thank the reviewer for these supportive comments. We have revised the manuscript taking all of the suggestions into account. Please find our point-to-point responses to the comments below.

1, The authors have performed immense of comparisons between cell ratios in KD patients before and after IVIG therapy by pairing statistics, and have described the changes in details. However, they did not execute all corresponding comparisons between health and KD patients, which might be useful for understanding KD pathogenesis.

To improve the comparisons between healthy controls and KD patients, we performed flow cytometric analysis for 20 additional healthy children. In the revised manuscript, all figures present the comparison between KD patients and healthy controls.

2, A recent published paper (Consiglio et al., 2020, Cell 183, 968–981) also investigated in the children immunology with COVID-19, KD and health controls. That study also described cell ratio changes and indicated that IL17A might drive Kawasaki disease. This study would be more convincing by compare and discuss their findings with that paper.

We discussed our findings with Consiglio et al. (Line 408-415). While they found difference in CD4 T cell subsets between KD and MIS-C, they did not find difference in CD4 T subsets between KD and healthy controls. This result is consistent with our data. Another point of that paper is, they found plasma IL17A level was significantly higher in KD than that in healthy controls. However, the expression level of *IL17A* in PBMCs is low (as shown in Fig. 3C), so we cannot detect differential expression of *IL17A* in our scRNA-seq data.

Minor suggestions:

1. The percentage number showed on the UMAP plot (Figure 1) makes it difficult to find specific cluster number on the UMAP plot. Instead, the percentage number could be moved to a supplemental table or figure. Try ratio-of-observe/expectation scores and plot a heatmap figure could simply and directly present the enrichment of cell type in patients and health controls.

Good suggestion. In the revised UMAP plot, we only preserved the percentage number of major cell compartments. The percentage of subtypes was moved to Supplementary Fig. 3A. We also present a heatmap of ratio-of-observed-to-expected percentage in Supplementary Fig. 3B.

2. In Figure 2B, the authors validated the percentage of lymphocyte compartments by FACS experiments on additional 16 KDs samples before and after treatment. Please illustrate the FACS results on normal controls and comparisons between KD and normal controls.

Now we have illustrated the flow cytometric data of 20 healthy controls and compared them with KD in Fig. 2B.

REVIEWERS' COMMENTS

Reviewer #1 (Remarks to the Author):

The authors have adequately addressed my comments.

I would suggest a minor edit at line 423 of the discussion:

"This study was performed on PBMCs and reflected the "systemic immune profiles", which might not be reflective of local inflammatory responses of immune cell infiltration in the coronary artery." and replace by the following:

"This study was performed on PBMCs and might not reflect the local inflammatory response developing in the coronary artery."

Reviewer #4 (Remarks to the Author):

I appreciate that the authors made efforts to address the issues, according to my previous comments. In the revised manuscript, instead of adding more samples of healthy control for single-cell analysis as suggested, they performed flow cytometric analysis for 20 additional healthy children and examined cell ratios between healthy controls and KD patients. I think this can make the results more convincing for analysis between KD (pre- and post-IVIG) and healthy controls. The authors have cited the recent published paper (Consiglio et al., 2020, Cell 183, 968–981) and compared findings between these two studies. I think this can improve the reasoning of discussion on KD pathogenesis. The authors have also addressed two minor issues which might help with the readability of this manuscript.

Overall, the analysis of this study is interesting and novel, and can be valuable for readers of nature communication with its insights on KD pathogenesis. Therefore, I recommend this manuscript for acceptance.

Responses to Reviewer #1:

The authors have adequately addressed my comments.

I would suggest a minor edit at line 423 of the discussion:

"This study was performed on PBMCs and reflected the "systemic immune profiles", which might not be reflective of local inflammatory responses of immune cell infiltration in the coronary artery." and replace by the following:

"This study was performed on PBMCs and might not reflect the local inflammatory response developing in the coronary artery."

The sentence was edited as suggested (Line 421).